# Non-Coding RNAs: Foes or Friends for Targeting Tumor Microenvironment

**DOI:** 10.3390/ncrna9050052

**Published:** 2023-08-28

**Authors:** Anna Szymanowska, Cristian Rodriguez-Aguayo, Gabriel Lopez-Berestein, Paola Amero

**Affiliations:** 1Department of Experimental Therapeutics, The University of Texas MD Anderson Cancer Center, Houston, TX 77054, USA; aszymanowska@mdanderson.org (A.S.); crodriguez2@mdanderson.org (C.R.-A.); glopez@mdanderson.org (G.L.-B.); 2Center for RNA Interference and Non-Coding RNA, Department of Cancer Biology, The University of Texas MD Anderson Cancer Center, Houston, TX 77054, USA

**Keywords:** non-coding RNAs, cancer, tumor microenvironment, immune response, anticancer therapy

## Abstract

Non-coding RNAs (ncRNAs) are a group of molecules critical for cell development and growth regulation. They are key regulators of important cellular pathways in the tumor microenvironment. To analyze ncRNAs in the tumor microenvironment, the use of RNA sequencing technology has revolutionized the field. The advancement of this technique has broadened our understanding of the molecular biology of cancer, presenting abundant possibilities for the exploration of novel biomarkers for cancer treatment. In this review, we will summarize recent achievements in understanding the complex role of ncRNA in the tumor microenvironment, we will report the latest studies on the tumor microenvironment using RNA sequencing, and we will discuss the potential use of ncRNAs as therapeutics for the treatment of cancer.

## 1. Introduction

Carcinogenesis is induced by changes in cellular, genetic, and epigenetic levels. Recent reports of the cancer genome have identified hundreds of thousands of mutations in coding sequences that promote cancer development. However, because coding sequences make up only 2% of the genome, scientists are also studying the functions of non-coding sequences in cancer transformation [1]. Non-coding RNAs (ncRNAs) are nucleic acids that do not code proteins. Based on cellular function, two types of non-coding RNAs can be distinguished: housekeeping non-coding RNAs (rRNA (ribosomal RNA), tRNA (transfer RNA), snRNA (small nuclear RNA), snoRNA (small nucleolar RNA), TERC (telomerase RNA), tRF (tRNA derived fragments)) and regulatory non-coding RNAs (miRNA (micro RNA), siRNA (small interfering RNA), piRNA (PIWI-interacting RNA), eRNA (enhancer RNA), lncRNA (long non-coding RNA), circRNA (circular RNA)) [2,3]. Regulatory non-coding RNAs are crucial for controlling cancer pathogenesis via regulating the processes of transcription, translation, and gene activity in cancer pathogenesis [4]. In addition, they also modulate immune cells’ response to tumors (e.g., circARSP91 activates the cytotoxic activity of natural killer (NK) cells in hepatocellular carcinoma) [5], epithelial-mesenchymal transformation (EMT) (e.g., lncRNA VIMAS1 promotes EMT in stomach cancer via Wnt/β-catenin signaling) [6], metastasis (piRNA-54265 promotes invasiveness of colorectal cancer) [7], and tumor angiogenesis (miRNA-126 inhibits the formation of new tumor blood vessels by blocking VEGF signaling in breast cancer) [8]. There are many methods for analyzing non-coding RNAs, including RT-qPCR, microarrays, bulk RNA-seq (bulk RNA sequencing), sc-RNA-seq (single-cell RNA sequencing), sp-RNA-seq (spatial RNA sequencing), metabolic labeling, chromatin immunoprecipitation (ChIP), and nuclear run-on (Table 1) [9,10,11,12,13,14,15,16].

Among them, single-cell and spatial RNA sequencing techniques provide the most detailed information about tumor cells and the surrounding environment (referred to as the tumor microenvironment, TME). It is known that cancer cells affect adjacent cells to promote their proliferation, invasion, and migration [9,10,12,13,14,15,16,17]. TME is heterogeneous and consists of cancer cells, cancer-associated fibroblasts, fibroblasts, pericytes, endothelial cells, immune cells (B cells, neutrophils, dendritic cells, T cells, NK cells, macrophages), mesenchymal cells, stromal cells, myofibroblasts, and epithelial cells. Cancer cells can specifically reprogram healthy surrounding cells into cells that promote tumor growth [17]. Single-cell and spatial RNA sequencing analyses of cells in the TME may enable the design of therapies that can affect cancer cells, the TME, or both. In this context, non-coding RNA might be a promising tool to develop targeted cancer therapy and/or the TME to induce a cytotoxic immune response against cancer (e.g., miRNA-149-3p downregulates expression of inhibitor receptors and promotes T-cell proliferation) [18], induce apoptosis in cancer cells (e.g., LOC285194 siRNA leads to activation of an external apoptosis pathway) [19], or restore drug sensitivity (miRNA-21 increases the sensitivity of glioblastoma cells to paclitaxel) [20] (Figure 1).

Exploring the role of non-coding RNA in cancer development may lead to the design of new anticancer therapies. Numerous clinical trials have been performed to deactivate gene expression using siRNA and thus inhibit the proliferation of cancer cells [21,22]. Despite the promising anticancer activity of non-coding RNA, the short half-life, low efficiency of encapsulation, and toxicity of these systems make it challenging to implement them in the clinical arena [23]. Therefore, in this review, we will discuss the role of non-coding RNA in TME and the latest achievements in RNA-sequencing in various cancers to identify potential targets for anticancer treatment based on non-coding RNA.

**Figure 1 ncrna-09-00052-f001:**
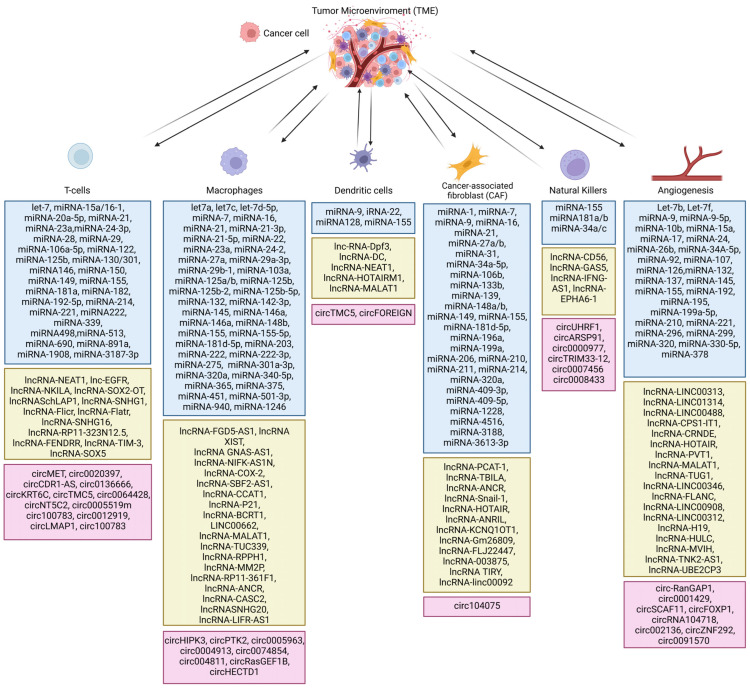
Types of non-coding RNA in the tumor microenvironment. Blue boxes show miRNAs, green boxes show long noncoding RNAs and pink boxes show circular RNAs [24,25,26,27,28,29,30,31,32,33,34,35,36].

## 2. Role of Non-Coding RNAs in TME

Cancer creates and expands the TME, which in turn supports cancer progression. It is hypothesized that the development of cancer is driven by mutations and by changes in cell homeostasis. There are two types of genetic mutations: germinal (hereditary mutations) and somatic (non-hereditary mutations) [37]. It has been demonstrated that hereditary breast and ovarian cancer predisposition syndrome (HBOC) is mainly caused by germline mutation of BRCA1/2 genes. Testing BCRA1/2 mutations is expensive and time-consuming. Therefore, there is an ongoing search for novel techniques that can provide quick and affordable diagnoses. In 2020, Marques et al. attempted to analyze the expression of miRNAs as potential biomarkers to detect hereditary breast cancer. Researchers found out that in sporadic breast cancer, forty-nine miRNAs have increased expression compared to normal breast tissue, while in the case of hereditary breast cancer, seventy miRNAs had increased expression. Among these miRNAs, eight were differentially expressed in sporadic and hereditary breast cancer. These findings provide one of the first proofs of the usage of miRNAs as biomarkers to detect hereditary breast cancer [38].

Somatic mutations may also occur in miRNA genes in various cancers. In 2020, Kozłowski et al. performed the first thorough analysis of somatic mutations in these ncRNAs. The percentage of mutations in the miRNA genes varied according to the type of cancer. Cancer SKCM (skin cutaneous melanoma) exhibited the highest number of miRNA mutations and THCA (thyroid carcinoma) had the lowest. Moreover, over eighty miRNA coding genes were mutated in PAN-Cancer. This discovery may help to identify target miRNA genes for cancer treatment [39]. The miRNA analysis of the human genome revealed that there are more germline mutations in miRNA than somatic mutations. The specific role of each mutation in cancer development is still unknown. It has been discovered that mutations in miRNA-16-1-15α in 13q14,3 intron 4 of *DLEU2* may be a predisposition for the development of CLL and breast cancer [40].

In 2012, Ziebarth et al. determined how somatic mutations of 3′UTR affect miRNAs in cancer. It has been demonstrated that somatic mutations of five genes (*TAL1*, *BMPR1B*, *KDM5A*, *SCG3*, *BCAS3*) disrupt the binding of miRNA and increase the expression of these genes which are prominent in cancer development [41].

Furthermore, both mutations, somatic and germline, of the 3′UTR *KLK*3 gene interfere with the binding of miRNA-675, miRNA-138, and miRNA-210, which promote the expression of the *KLK3* gene in prostate cancer and regulate cancer proliferation. The identification of the disturbance caused by germline and/or somatic mutations may help identify prominent miRNA targets for cancer treatment [41].

Uncontrolled tumor growth accelerated cancer cell metabolism, and chaotic vascularization of cancer cells leads to hypoxia in the TME. The communication among the TME cell types is in part mediated by secreted proteins, cytokines, and non-coding RNAs transported by exosomes [42]. These factors in TME may induce epithelial-mesenchymal transition which increases the invasiveness of cancers [43].

### 2.1. Cytokines and Non-Coding RNA Intertwined in TME

Cytokines have a significant role in the communications of cancer cells and other TME cells [44]. Cytokines are proteins that regulate the proliferation, growth, and activation of immune cells [45]. Categorized according to their function, cytokines include chemokines (which induce chemotactic migration) [46], interleukins (involved in immune and inflammation processes) [47], interferons (which stimulate an immune response) [48], and tumor necrosis factors (TNFs) (pro-inflammatory proteins) [49].

Chemokines can be divided into four groups according to their structure: CC, CVC, CX3C, and XC [50]. One of the chemokines that induce cancer progression via the PI3K/AKT, JAK/STAT3, MAPK/ERK, and NF-κB pathways is CCL-5. CCL-5 produced by cancer cells induces cancer cell proliferation but also evokes infiltration of T cells and dendritic cells (DCs) into the TME. Overexpression of CCL-5 is observed in several solid cancers and leukemias [51]. Recently, Chen et al. showed that the expression of CCL5 in oral squamous cell carcinoma is modulated by lncRNA *ZFAS1* which binds miRNA-6499-3p. The histological analysis revealed that lncRNA *ZFAS1* is upregulated in tumor tissue. Moreover, in vitro studies showed that silencing lncRNA *ZFAS1* using siRNA inhibits cancer progression, colony formation, invasion, and migration. The oncogenic character of lncRNA *ZFAS1* is correlated with the downregulation of miRNA-6499-3p, which leads to increased expression of CCL-5. These findings identify lncRNA *ZFAS1* as a potential target for anticancer treatment [52].

The second essential group of cytokines associated with TME is interleukins. More than 40 interleukins may be involved in tumorigenesis. They have a dual role in cancer progression: (1) recruiting anticancer cells to the TME (by IL-1β) and (2) suppressing anticancer immune response by promotion of T cell inhibitory receptors including PD-1, LAG3, TIM-3, and TIGIT (by IL-35) [47]. The diverse properties of interleukins make them a target of interest for anticancer therapies. The U.S. Food and Drug Administration (FDA) approved the first drug containing an interleukin (interleukin-2) for metastatic renal cell carcinoma treatment almost 30 years ago [47]. However, IL-2 causes serious adverse effects including cardiac and pulmonary toxicity [53]. Currently, there are 2072 clinical studies to use not only interleukin-2 in cancer treatment but also IL-12, -15, -18, -11, -6, -4, -7, and -21 [54]. Among them, it is worth noting that IL-6 is overexpressed in solid tumors and multiple myeloma [55]. IL-6 promotes tumor development by activation of the PI3K/Akt, NF-kB, and Mek/ERK pathways [56]. Thus, Kishimoto et al. showed that IL-6 increases the expression of lncRNA AU021063 which activates the Mek/ERK pathway in breast cancer. Researchers demonstrated that inhibition of any of these molecules may be important to develop novel strategies for breast cancer treatment [57].

Interferons may also play a key role in the TME, as they are secreted by immune and tumor cells. Interferons can modify immune activity against malignant diseases [48]. There are two types of interferons: –I and –II. These two groups play several functions in carcinogenesis including the promotion of production of IL-12 by DCs (IFN I), inducing cytotoxic activity of NK cells (IFN II), inducing transformation of macrophages from M1 to M2 (IFN I and II), and increasing the cytotoxic function of T cells (IFN I) [48,58]. Among these two groups, IFN-γ plays a key role in immune antitumor response [59]. In 2020, Chiocca et al. showed that IFN-γ signaling in glioblastoma is correlated with the expression of one of the non-coding RNAs: lncRNA *INCR1*. The bulk RNA sequencing analysis of patient-derived glioblastoma cell lines activated by IFN-γ revealed that lncRNA *INCR1* was the most upregulated non-coding RNA. It has been shown that cell lines with high expression of this non-coding RNA also had a high expression of PD-L1. Knockdown of lncRNA *INCR1* in the glioblastoma cell line led to inhibition of IFN-γ, and PD-L1 which allows the cytotoxic activity of T cells in vivo. In conclusion, lncRNA *INCR1* may be a promising target for anticancer therapies because it plays a significant role in the regulation of IFN-γ signaling and immune antitumor response [60].

TNF-α is involved in a variety of regulatory processes in normal and cancer cells. There are two types of TNF-α receptors: TNFR1 (occurring in all cell types) and TNFR2 (mostly found in immune cells). It has been demonstrated that activation of TNFR2 leads to the progression of cancer. However, a series of preliminary studies have demonstrated that, depending on the cancer type, TNF-α can promote or inhibit apoptosis in tumor cells [61]. The anticancer activity of TNF-α is also associated with non-coding RNA, especially miRNA-145 [61]. Overexpression of miRNA-145 in combination with TNF-α leads to the induction of apoptosis in MDA-MB-231 cell lines via an external pathway [62]. Moreover, in cervical cancer, miRNA-130a downregulates TNF-α expression while TNF-α decreases the expression of miRNA-130a by activation of NF-κB [63].

### 2.2. Exosomes as External and Internal Carriers of Non-Coding RNAs in TME

Exosomes are messengers involved in the communication network in the TME [64]. Exosomes are small vesicles (30–150 nm) produced by various cells, including cancer and TME cells. Exosomes may contain a wide variety of cytoplasmic contents including non-coding RNA, integrins, lipids, and enzymes such as matrix metalloproteinase (MMP), among others.

Exosomes have unique properties such as high stability, low risk of induction of immune response, and the capability to cross biological barriers [65]. These natural extracellular molecules may be produced by various cell types of animals and plants and used as nanocarriers [65]. In 2020, Tao et al. loaded Bcl-2 siRNA in exosomes isolated from bovine milk. The in vitro results revealed that the designed system enabled the transfection of pancreatic and colon cancer cells with high efficiency. Moreover, this system decreased the survival of cancer cells by induction of apoptosis. The in vivo studies confirmed the anticancer activity of this approach system [66]. However, exosomes are isolated from fresh cells, and large-scale production may be cost-inefficient. The process of loading ncRNAs into exosomes is inefficient, and even loaded ncRNAs may form aggregates, which decreases the loading yield [65].

Exosomes are also produced in TME. Cancer-derived exosomes packed with ncRNAs may regulate angiogenesis, metastasis, cancer proliferation, and migration and modulate the immune system [67,68,69,70]. In 2019, He et al. showed that exosomal miRNA-499a-5p induces cancer proliferation and invasion by the promotion of EMT, activation of the mTOR pathway, and inhibition of apoptosis in non–small cell lung cancer (NSCLC) [68].

In addition, exosomes secreted by the TME may induce resistance to chemotherapy, e.g., exosomes packed with lncRNA-SNHG14 associated with the promotion of resistance to trastuzumab in breast cancer patients with HER2 (+), lncRNA-ARSR linked with resistance to sunitinib in renal cancer cells, lncRNA-SBF2-AS1 promotes resistance to temozolomide in glioblastoma cells, and lncRNA-CCAL promotes resistance to oxaliplatin in colorectal cancer cells. These suggest that exosomes might be crucial targets to overcome chemotherapy resistant [71,72].

Recent studies showed that exosomal lncRNAs also take part in the regulation of autophagy in the TME. lncRNA ANRIL promotes autophagy by preventing miRNA-99a and miRNA-449a binding to beclin 1 [72,73]. Overexpression of this lncRNA is observed in the urine of bone cancer patients. The results of this study may allow in the future to develop a new diagnostic method for bone cancer using lncRNA ANRIL [72]. The differences between the level of exosomes packed with ncRNAs in normal and cancerous cells make them potential biomarkers. It has been demonstrated that lncRNA H19 may be considered a predictive breast cancer biomarker. In prostate cancer, it has been shown that lncRNA-SAP30L-AS1 and SChLAP1 are important biomarkers. In NSCLC, lncRNA-GAS5 may be considered a diagnostic marker in the early stages of cancer development [71].

In 2016, Kanlikilicer et al. showed that exosomes derived from sensitive ovarian cancer cells (HeyA8, SKOV3-ip1, A2780) and resistant ovarian cancer cells (HeyA8-MDR, SKOV3-TR, A2780-CP20) have upregulated expression of miRNA-6126. Following this discovery, the most significant change in the expression of miRNA-6126 was visible in HeyA8 ovarian cancer cells. In the next step, the researchers analyzed the pathways that are regulated by miRNA-6126. It has been presented that overexpression of miRNA-6126 leads to reduced activation of the PI3K/AKT pathway in HeyA8-MDR, SKOV3-ip1, and SKOV3-TR cells, which leads to decreased invasion and migration of ovarian cancer cells. The in vivo studies confirmed the suppressor activity of miRNA-6126. These studies provide evidence of tumor suppressor activity of miRNA-6126 in ovarian cancer treatment [74]. In 2018, Lopez et al. demonstrated that miRNA-1246 is highly overexpressed in ovarian cancer cells (A2780, A2780-CP20, HeyA8m HeyA8-MDR, SKOV3ip1, and SKOV3-TR) compared to normal cell line HIO180. Researchers presented that miRNA-1246 plays an important role in the Cav1/PDGFRβ pathway. In the in vivo studies, combination therapy using an inhibitor of miRNA-1246 and paclitaxel led to the significant inhibition of tumor growth, which was associated with reduced proliferation of cancer cells, downregulation of PDGFRβ, and upregulation of Cav1 in cancer tissue. It has been presented that exosomal miRNA-1246 secreted by ovarian cancer cells SKOV3-ip1 leads to an increase in miRNA-1246 in M2-type macrophages. This shows that exosomal miRNA-1246 secreted by ovarian cancer cells my affect TME to promote the progression of cancer [75], thus, these two elegant manuscripts illustrate the role of exosomal miRNA in intercellular communications in the TME.

Exosomes containing ncRNAs may also affect immune cells. They can promote the M2 phenotype of macrophages (lnc-RPPH1 in colon cancer, lnc-BCRT1 in breast cancer), inhibit differentiation of T cells (miRNA-24-3p in nasopharyngeal carcinoma), impair T cell function (circRNA-002178 increase PD-1 expression in lung adenocarcinoma), and inhibit the immune response of dendritic cells (miRNA-203 in pancreatic cancer). Therefore, determining the molecular profile of exosomes found in the plasma of cancer patients may have the potential to assess the immune status of the patient and predict their response to immunotherapy [76].

The development of cancer tumors involves the supplementation of essential nutrients and oxygen to TME. CircRNAs including circRNA-SHKBP1, circRNA-100338, circR-NA-0007334, circRNA-KIF18A, circRNA-29, and circRNA-HIPK3 encapsulated in exosomes play an important role in this process [77]. The mechanism of regulation of angiogenesis by exosomal circRNAs is based on binding targeted miRNAs associated with tumor angiogenesis. These findings may be fundamental for the development of antiangiogenic cancer therapies in the future [78].

### 2.3. Role of Non-Coding RNA in Cancer-Associated Fibroblasts (CAFs) in TME

Another important component of the TME is cancer-associated fibroblasts (CAFs). There are five types of CAFs: the F1 group inhibits tumor growth, F2 stimulates tumor progression, F3 affects angiogenesis and tumor immunity, F4 regulates the transformation of the extracellular matrix, and the function of the F5 group is still unknown [79]. Contrary to normal fibroblasts, CAFs express α-SMA, FSP1, FAP, NG2, and PDGFRα/β [80], which promote cancer growth, enhance angiogenesis, and contribute to intercellular matrix remodeling. It was shown that increasing the expression of miRNA-31 in the co-culture of CAF and EC-1 reduces the invasion and migration of EC1 cells without effect on their proliferation [81,82]. Since then, around 16 non-coding RNAs in CAF have been described. It is worth noting that growth factors (FGF, PDGF) and hypoxia may activate fibroblasts to CAFs, which produce pro-cancer cytokines such as CXCL-1, -2, -3, -12, and -14; CCL-2, -5, and -17; and IL-18 [83,84,85]. In 2017, Zhao et al. reported that CXCL-14 secreted by CAFs in ovarian cancer can increase the level of lncRNA-*LINC00092*, which results in a higher risk of metastasis. These results show that the inhibition of CXCL-14 or lncRNA-*LINC00092* may be a promising target for ovarian treatment [85].

### 2.4. Non-Coding RNA in the Regulation of Response of Macrophages in TME

Macrophages are crucial in cancer progression. Moreover, they constitute more than 50% of the TME. These cells regulate inflammation and early carcinogenesis. Cancer cells secrete chemokines (MCP-1, MIP-1α, VEGF, CSF1R) that favor the infiltration of monocytes, which are precursors of macrophages [86]. The monocytes in the TME differentiate under the influence of IL-6 and CSF-1 into macrophages, specifically tumor-associated macrophages (TAMs). In contrast to normal macrophages, TAMs do not have an immunostimulant function. TAMs that have an M2-like phenotype produce anti-inflammatory cytokines (IL-4, IL-10, IL-13), factors stimulating the formation of blood vessels (VEGF, IL-1β, TNF-α, IL-8, PDGF-β), growth factors (EGF, IL-6, bFGF), and cell migration factors (metalloproteinases 2, 7, 9). These unique qualities of TAMs promote tumor development by induction of angiogenesis, expression of oncogenes, and anti-apoptotic signaling in the TME [86,87]. The diverse functions of TAMs make them a promising target of anticancer therapy. There are two approaches to utilizing TAMs in cancer treatment: inhibition of the recruitment of TAM cells to the TME and restoration of antitumor activity of TAMs. Relationship between Macrophages and Non-Coding RNA in TMThe first anticancer strategy utilizes ncRNA to inhibit the recruitment of TAM to TME. The process of TAM survival and migration to the TME is controlled by CSF1, whose induced expression is linked with a poor prognosis of breast, prostate, endometrial, bladder, kidney, and esophageal cancer patients. Blockage of CSF1 expression by siRNA in the breast cancer in vivo model inhibited tumor growth and reduced the recruitment of macrophages to TME [88,89]. Olayioye et al. showed that the expression of CSF1 is linked with miRNA-149 expression in breast cancer. Upregulation of miRNA-149 in MDA-MB-231 cells leads to inhibition of recruitment of TAM to TME in vitro and in vivo. miRNA-149 also impedes M2 polarization [90]. The second approach utilizing TAMs to treat cancer is based on restoration of the antitumor activity of macrophages, which may be achieved through IgSF (immunoglobulin superfamily) inhibitors, immune checkpoint inhibitors, PI3Kγ inhibitors, agonists of Toll-like receptors (TLRs) or CD40, and non-coding RNA (siRNA and miRNA) [91]. In 2018, Yin et al. designed siRNA to target VEGF and PIGF. These two markers of angiogenesis are overexpressed in M2-type macrophages and breast cancer cells and the use of siRNAs against them led to the reversion of the M2 to M1 phenotype of TAM in TME which led to the inhibition of tumor growth in vitro and in vivo [92]. Moreover, because macrophages make up the majority of the TME, they are an interesting system for delivering non-coding RNA into cancer cells. In 2019, Wayne et al. developed an efficient method of loading siRNA lipoplex into macrophages. This modified macrophage system was efficient in transporting siRNA to cancer cells. When they tested the anticancer activity of the designed system, they found that macrophages loaded with CIB1 siRNA and co-cultured with breast cancer cells decreased the viability of the cancer cells [93].

### 2.5. Non-Coding RNAs in the Regulation of T-Cell Activity in the TME

In the TME, three subtypes of T cells can be distinguished: naive (mature T cells, precursors for other T cell subsets), memory (T cells that can recognize specific antigens), and effector (which have various functions in the immune response and includes cytotoxic, helper, and regulatory T cells) [94,95,96]. The functions of T cells in the TME are still largely unknown. On one hand, T cells have anticancer activity: CD8^+^ T cells decrease the viability of cancer cells, and Th1 cells secrete pro-inflammatory factors like TNF-α and IFN-γ, which stimulate the immune system against cancer cells. However, T cells also have pro-tumor activity; CD4^+^ T cells inhibit CD8^+^ T cells’ anticancer activity by producing IL-4 and -10, and Th17 cells produce IL-17, which promotes tumor growth. In addition, CD1^+^ TIM3^+^ T cells are exhausted and not able to mount an anticancer response [94,95,96].

In the TME, CD4^+^ and CD8^+^ T cells are inactivated by increased expression of immune checkpoints including PD-1, CTLA-4, LAG-3, TIM-3, BTLA, and TIGIT, which leads to anergy and exhaustion of T cells. Exhausted T cells have a high expression of CD69 and CD44 and a low expression of CD62L and are unable to secrete IL-2, TNF-α, IFN-γ, and granzyme B, which play crucial roles in anticancer activity. Therefore, reversion of T-cell exhaustion is an approach of interest for treating cancer [97]. Investigators have found that knocking down PD-1 in T cells using siRNA in combination with knocking down PD-L1 in the MCF-7 cell line can restore the effector function of lymphocytes and significantly decrease cancer cell survival. Moreover, the knockdown of PD-1 and PD-L1 increased IFN-γ and TNF-α production in vitro [98]. Deregulation of T-cell immune response in the TME may be associated with ZFP91, an oncogene protein that induces cancer progression and invasion and inhibits programmed cell death. Wang et al. knocked down ZFP91 in T cells in colon adenocarcinoma, which induced T-cell proliferation and anticancer activity. The knockdown of ZFP91 also increased the expression of genes linked with glucose metabolism, which are crucial for the anticancer activity of T cells. Furthermore, the knockdown of ZFP91 decreased the activation of the mTORC1 pathway, which is important in T-cell glycolysis [99].

### 2.6. Non-Coding RNAs in the Regulation of B Cells in TME

As mentioned, the TME is composed largely of immune cells, including B cells. The role of B cells in carcinogenesis has been underestimated for many years. Nonetheless, recent data suggest that B cells in the TME might be crucial in the anticancer response. In 2021, Lopez-Berestein et al. using Explainable Artificial Intelligence (XAI) showed that B cells are one of the most critical elements of TME for the prognosis of breast cancer. High expression of naïve B cells in TME was associated with better overall survival of breast cancer patients [100]. Also, Xia et al. showed that B cells play an important role in colon cancer. It has been demonstrated that one of the regulators of B cells in the TME is CXCL-13, which activates its receptor CXCR5. This receptor influences cancer cell differentiation (via miRNA-23a), migration (via MMP, N-cadherin, E-cadherin, Slug/Snail), invasion (via FAK/ERK, c-Myc/RANKL), and proliferation (via DOCK2/JNK) [101,102,103].

### 2.7. Non-Coding RNA in the Regulation of EMT in TME

EMT (epithelial-mesenchymal transition) is a crucial process in the development of metastasis and tumor progression. In EMT, a cell is deprived of connectivity with other cells and acquires the capacity to migrate. EMT is stimulated by increased expression of vimentin, SNAI1/2, TWIST, ZEB1/2, and non-coding RNAs including miRNA-200 and miRNA-205 [43]. In 2019, it was demonstrated that tamoxifen-resistant breast cancer cells express more EMT markers compared to non-resistant breast cancer cells. Moreover, tamoxifen resistance was associated with decreased expression of miRNA-200b and miRNA-200c. Transfection of resistant cells with pre-miRNA-200b and -200c notably reduced the expression of c-MYB. This finding prompted researchers to silence c-MYB, which restored the sensitivity of breast cancer cells to tamoxifen and reduced EMT [104]. The EMT process is controlled by various epigenetic mechanisms including DNA methylation and histone modifications [105]. In 2021, Zheng et al. used lovastatin (a cholesterol-lowering drug) against breast cancer. Lovastatin was shown to deregulate lysine succinylation of eight proteins, whereas elevated expression of these proteins in triple-negative breast cancer was linked with poorer overall survival. The mechanism of the anticancer activity of lovastatin was also associated with the inhibition of EMT in vitro and in vivo. Lovastatin also decreased metastasis to the liver [106].

Chemotherapeutics that act directly on cancer cells have limitations that include drug resistance as well as physiological and anatomical barriers to delivery to cancer cells. Cancer cells are genetically unstable, so some molecular targets that make cancer cells chemosensitive may be eliminated during treatment, leaving only cells resistant to treatment. These changes require a constant search for novel targets. As an alternative, drugs targeting the TME can affect growth factors, cytokines, and immune cells that are engaged in the interaction between normal and cancer cells. Despite the limitations of both types of drugs, the goal of current research is to find drug combinations that both act on cancer cells and modify the TME.

## 3. RNA Sequencing to Characterize ncRNAs in the TME

Understanding the interactions between cancer cells and cells in the TME may enable the development of a therapy that recognizes specific molecular targets. One of the innovative methods to characterize TME is RNA sequencing, which helps to analyze prominent ncRNAs in cancer development [107]. Identification of new molecules involved in the pathogenesis of disease is critical for the development of new drugs. There is an increased interest in human genetics in drug development, suggested by the increase in genomic research studies. However, emerging challenges provide a complex scenario. The generation of new targeted delivery systems may help in resolving those challenges and opening the way to exploit the potential of human genetic analysis for the development of new drugs [108]. RNA sequencing is a promising tool for the precise determination of gene expression. Currently, there are three RNA sequencing technologies on the market: bulk RNA-seq, single-cell RNA-seq, and spatial RNA-seq. The first is based on measuring the average gene expression in a cell population, not the distinct cells that compose the tumor. However, this analysis allows for the identification of genetic and transcriptional heterogeneity between patients with the same cancer and their response to therapy [109]. Moreover, the elaboration of computational deconvolution methods allows for the analysis of different cell types in bulk RNA sequencing. Depending on the chosen deconvolution method, researchers can obtain information about cell type fraction and cell types [110]. Single-cell RNA-seq allows for the measurement of the level of gene expression between cell populations, and the third, the most advanced, allows for the quantitative analysis of gene expression and visualization of its distribution on tissue sections [13].

In 2013, 2017, and 2021, the American College of Medical Genetics and Genomics (ACMG) published a list of secondary findings associated with exome and whole genome sequencing. In these reports, 73 variants of genes are recommended to cause disease [111,112]. However, identical genetic variants can have different symptoms in different patients. Therefore, it is extremely important to study the penetrance, pleiotropy, and expressivity of various gene variants **[113]**. It is worth noting that around 73% of clinical projects with genetic linkage to the disease were active/successful in Phase 2 [114].

### 3.1. RNA Sequencing for Breast Cancer

Breast cancer is the most frequently diagnosed cancer in the world, as well as the most common cause of death in women from cancer [115]. Breast cancer is a highly heterogeneous group in which therapy is chosen depending on the molecular subtype of the tumor [116]. However, even within the same histological subtype, response to therapy can vary. In 2022, Yang et al. identified crucial lncRNAs for luminal A breast cancer using the RNA sequencing method. It has been demonstrated that 151 lncRNAs are upregulated and 121 lncRNAs are downregulated in luminal A breast cancer compared to normal tissue. It has been demonstrated that four of these lncRNAs in complex with mRNA may be involved in the development and progression of luminal A breast cancer (OXC-AS3-HOXC10, AC020907.2-FXYD1, AC026461.1-MT1X, and AC132217.1-IGF2). The results provide a basis for future RNA sequencing studies on a larger group of patients [117]. Other scientific reports noted that various cancer subtypes have different percentages of tumor-infiltrating lymphocytes [118]. The highest proportion of tumor-infiltrating lymphocytes was observed in triple-negative breast cancer samples, while the lowest was in luminal HER2^−^ cancers. Variations like these in the immune TME makeup across breast cancers suggest that immunotherapy may be more or less effective according to the composition of the TME in different breast cancer subtypes [118].

### 3.2. RNA Sequencing for Lung Cancer

In 2020, Deng et al. using RNA-seq TCGA, TAGA, and Aligent datasets discovered that twenty lncRNAs have different expressions in lung cancer compared to normal tissue [119]. Researchers validated the results obtained from TCGA and Aligent datasets, which showed that eight lncRNAs are present in both groups. These eight lncRNAs may be used in the future as potential biomarkers for NSCLC (non-small cell lung cancer) diagnosis. Moreover, high expression of one of the selected lncRNAs, LINC02555, is associated with poor patient survival rate [119].

Preliminary single-cell RNA sequencing analysis showed that the TME in advanced NSCLC includes cancer cells, myeloid cells, fibroblasts, T cells, B cells, endothelial cells, epithelial cells, mast cells, alveolar cells, neutrophils, and follicular DCs [120]. The highest number of T cells and B cells was observed in the TME of LUSC patients. CD4^+^ and CD8^+^ T cells had an elevated expression of immune checkpoints CTLA-4 and TIGIT. In the LUAD subtype, macrophages—rather than tumor cells as previously thought—inhibit T-cell function through interaction with inhibitory checkpoints (primarily TIGIT) [120].

### 3.3. RNA Sequencing of Colorectal Cancer

In the TME of colorectal cancer, five regions can be distinguished: smooth muscles, enterocytes, lamina propria, fibroblasts, and tumor cells. In these regions, researchers identified 12 different cell types based on the gene markers, with CAFs representing a high proportion. There are two types of CAFs in colorectal cancer: inflammatory CAFS and myo-CAFs. Inflammatory CAFs are involved in multiple synaptic pathways including EMT, cholesterol homeostasis, and metabolism of bile and fatty acid, while myo-CAFs rearrange the extracellular matrix [121]. In 2018, Goel et al. using bulk RNA sequencing identified seventy-two deregulated lncRNAs in colorectal cancer patient samples. However, only forty-nine genes were also affirmed by TCGA results. In vitro studies, researchers showed that the silencing of two lncRNAs upregulated in colorectal cancer (CRCAL-3 and -4) led to the inhibition of cell proliferation and formation of colonies. The effect was more visible in cells transfected with siRNA against CRCAL-3. This shows the potential of RNA sequencing in discovering novel targets for colorectal cancer targeted strategies [122].

### 3.4. RNA Sequencing of Ovarian Cancer

While ovarian cancer is a tumor of heterogeneous histological origin, its most prevalent subtype is high-grade serous ovarian cancer (HGSC), which also shows the worst prognosis. High genetic variability within the same histologic type results in different responses to chemotherapy; hence, understanding the mechanisms responsible for resistance or sensitivity to therapy is an important aim. Anke van den Berg et al. using the RNA sequencing method of HGSC patients showed that seventy-nine miRNAs have significantly different expression compared to normal tubal tissue. Moreover, it has been demonstrated that miR-145-5p is the most prominent miRNA associated with an increase in the tumor stage [123]. In 2022, Stur et al. presented a spatial RNA sequencing of 12 HGSC samples. In the TME of HGSC, they identified 20 types of cells including B cells, neutrophils, dendritic cells, myocytes, T-cells, CAFs, fibroblasts, NK cells, macrophages, mesenchymal cells, endothelial cells, myofibroblasts, EMT-like cells, monocytes, epithelial cells, pDCs (plasmacytoid dendritic cells), mature DCs, pBs (peripheral blood cells), stem cells, and conventional type 1 DCs. In the next step of the research, it was discovered that the expression of six genes differs between individuals who respond poorly to chemotherapy and those who respond excellently. This suggests that the expression of these genes may play a crucial role in determining the response to chemotherapy. The presented results aid in understanding how the tumor microenvironment (TME) influences the response to chemotherapy [124].

### 3.5. RNA Sequencing of Prostate Cancer

In 2020, prostate cancer was the second most common cancer in men. More than half of prostate cancer cases have an ERG-TMPRSS2 (ETS-related gene—Transmembrane Protease Serine 2) fusion on chromosome 21. This leads to overexpression of the oncogene ERG, which is involved in cell division, survival, cell differentiation, and apoptosis. The single-cell RNA sequencing of the immune composition of TME revealed differences in CD4^+^ T cell proportion in prostate cancer types based on the presence of ERG-TMPRSS2 fusion. Tumor cells with fused genes showed overexpression of FOSB, FOS, and JUN, while tumor cells without fused genes overexpressed CXCR6 and DUSP4 [125,126]. The transcriptome sequencing of prostate cancer cell lines, nonmalignant growth of tissue next to the prostate, localized prostate cancer, and metastatic prostate cancer revealed that more than 120 ncRNAs in prostate cancer may be significant for oncogenesis [127]. This shows the prominent role of ncRNAs in prostate cancer development and potential in designing future targeted therapies.

### 3.6. RNA Sequencing of Gastric Cancer

In the TME of gastric cancer, expected cancer cells can be distinguished as fibroblasts, lymphoid cells (T cells and NK cells), mast cells, B cells, dendritic cells, macrophages, epithelial cells, and endothelial cells. Comparative single-cell RNA sequencing of primary stomach tumor samples versus normal tissue revealed a lower proportion of epithelial cells and a higher proportion of myeloid cells in tumor samples. A higher percentage of plasma cells and a lower percentage of epithelial cells were noted in diffuse cancer in comparison to intratumoral gastric cancer. This may indicate that patients with diffuse gastric cancers have poorer prognoses [128,129,130]. In 2021, Akhavan-Niaki performed RNA sequencing on cancerous and non-cancerous samples derived from patients with early stages of gastric adenocarcinoma. The results showed that in tumoral samples, six ncRNAs are downregulated compared to normal tissue. A thorough analysis of the role of these ncRNAs in gastric cancer may allow us to understand the mechanisms responsible for cancer progression as well as develop effective gene therapies [131].

### 3.7. RNA Sequencing of Pediatric Cancers

The most common and fatal types of cancer in children are leukemias and central nervous cancers [132]. One of the main reasons for the high mortality rate of pediatric patients is the late diagnosis of the disease. Cancers occurring in children are mostly non-epithelial and their histological specificity highly depends on the age of the patient at the time of the onset of the disease, which makes them harder to diagnose in the early stage. Therefore, it is extremely important to evaluate cancer gene expression in adult and pediatric patients to understand the tumorigenesis process and design effective treatment [133].

#### 3.7.1. RNA Sequencing of Pediatric Leukemias

Leukemias represent about 25–33% of all cancers in children, which include ALL, AML, and CML [133]. There are two subtypes of ALL: T-cell acute lymphoblastic leukemia (T-ALL) originates from precursor T lymphocytes (thymocytes) maturing in the thymus, while the BCP-ALL (B-cell precursor ALL) subtype arises from B-cell precursors that undergo differentiation in the bone marrow. The frequency of both subtypes is ~15% and ~85%, respectively. In 2020, Bourque et al. performed a single-cell analysis of eight subtypes of leukemias in children. The comparison of cell types of bone marrow mononuclear cells of healthy adults and children showed that children had a lower percentage of T cells and NK cells and a higher percentage of erythrocytes, monocytes, and B cells compared to adults. These differences may be correlated with an inverse correlation of ribosomal protein expression in pediatric cancer cells. These findings in the future may enable the selection of genes responsible for cellular resistance to therapeutics in children [134]. The literature data indicate that relapse of ALL can be attributed to leukemic stem cells (LSCs) specifically. Therefore, in 2022, Lammens et al. assessed the expression of lncRNA and miRNA in acute leukemia (AML) using micro-array, small RNA-sequencing, and gene set enrichment analysis. It has been presented that the expression of 22 lncRNAs was substantially changed in leukemic stem cells compared to hematopoietic stem cells in pediatric AML samples. The findings from these studies offer new avenues for addressing pediatric AML through targeted treatments [135]. In the same year, Li et al. analyzed the expression of lncRNA and mRNA in B-ALL samples using the RNA sequencing method. It has been shown that fifteen lncRNAs and four mRNAs were downregulated in this disease compared to samples derived from patients with anemia and agrunolocytosis. It has been also demonstrated that five of these lncRNAs interact with mRNA and may play a prominent role in development of B-ALL in pediatric patients. This therapy aids in comprehending the impact of interactions between lncRNA and mRNA on tumor progression [136].

#### 3.7.2. RNA Sequencing of Pediatric Brain Tumors

CNS tumors are the second most prevalent type of childhood cancer after leukemias. They represent the most frequent type of solid tumors within this age range. Despite advancements in the field of neuro-oncology, they remain the primary cause of mortality among children with cancer [137]. It is worth noting that among malignant CNS tumors, the most common is medulloblastoma. In terms of molecular, demographic, histopathology, and prognostic, four subgroups can be distinguished: subgroups with activation of the WNT pathway activation (about 10%), subgroups with activation of the SHH pathway (approximately 25–30%), and groups 3 (about 25%) and 4 (around 35%). Each subgroup has inimitable molecular characteristic and anatomic changes, which requires adequate treatment [138]. In 2020, Perera et al. analyzed lncRNA expression in four medulloblastoma subgroups. Researchers identified eleven lncRNAs, whose expression varies depending on the subtype of medulloblastoma. Subsequently, scientists proceeded to divide medulloblastomas into subclasses based on the expression of these lncRNAs. In the analysis, they used patient-derived xenografts from three subtypes of medulloblastoma: SHH, group 3, and group 4. Unfortunately, at the time, WNT patient-derived xenografts were not available for analysis [139]. The research conducted by Perera et al. enables the identification of novel lncRNA targets for diagnosing and designing targeted therapy for medulloblastoma.

It is worth noting that despite the potential benefits of RNA sequencing analysis, the data obtained require computer processing, which involves the use of computer programs based on different algorithms. This means that the data obtained from different programs may vary [140].

## 4. ncRNA Promising Drugs and Drug Targets in Cancer Treatment

The discovery of the involvement of ncRNA in cancer development enriches the knowledge of the pathogenesis of basic cellular pathways and offers new targets for diagnosis and cancer therapy. One of the prerequisites for understanding the function of ncRNAs is to know the processes in which they participate and to identify the transcripts of genes and proteins that are involved in their regulation. There are two cancer treatment strategies based on ncRNA. The first approach involves using non-coding RNA to alter gene expression in cancer cells, while the second is based on inhibiting or increasing the expression of non-coding RNA in cancer cells using various strategies including antisense oligonucleotides (ASO) [141].

### 4.1. ncRNA-Based Therapeutics in Cancer Treatment

The discovery of the promising role of ncRNAs in cancer treatment still requires further research related to the stability of the ncRNA, delivery systems, and anatomical and physiological barriers as well as immunological response. One of the challenges is that ncRNAs are very sensitive to temperature. It has been demonstrated that the half-lives of miRNA, lncRNA, and circRNA were approximately 16 h, 17 h, and 24 h, respectively [142]. Modification of ncRNA with PEG (Polyethylene Glycol), non-anionic surface-active agents, or other chemical modifications may increase its half-life and effectiveness [143,144]. Modifications to ncRNAs can be introduced in the heterocyclic base (2,4-difluorotoluyl ribonucleoside; 2,6-diaminopurine; 5-bromoudirine; 5-iodouridine; 2-thiouridine; dihydrouridine; pseudouridine; 5-propynyluridine; 5-methyluridine; 5-methylcytidine), sugar ring (addition of group: 2;-fluoro, 2′-O-methylo; 2′-O-methoxyethyl; 2′-O-Allyl; 2′-O-(2,4-dinitrophenyl); 2′-deoxy-2′-fluoro-β-D-arabino; 2-deoxy; locked nucleic acid; 4′-thio; unlocked nucleic aicid; 2′-amimno; ethylene-bridge), or internucleotide bond modification (phosphodiester; phosphorothioate; boranophosphate; amide-linked; peptide nucleic acid; morpholino) [145]. The second obstacle is the efficient delivery of non-coding RNAs to targeted cells. The yield of ncRNA transport depends on the charge, lipophilic character, and stability of the carrier. It is recommendable that if the ncRNAs will be administrated systemically, the carrier should not have a positive charge because the cell membrane has a negative charge (−70 mV). Another obstacle to the use of ncRNA in cancer treatment is the possibility of activation of the immune response by the motif sequence in the ncRNA [143,146,147].

In conclusion, ncRNAs may be used as anticancer therapeutics because they are crucial regulators of the TME. ncRNAs are involved in the activation or inhibition of specific gene expression (Figure 2) [148,149,150,151,152].

#### 4.1.1. MiRNA-Based Therapeutics for Cancer Treatment

MiRNA regulates post-transcriptional gene expression (Figure 2A). It is known that over 2500 miRNAs modulate the expression of approximately 60% of human genes. miRNAs are classified as tumor suppressor miRNAs (inhibiting the expression of oncogenes or genes that induce apoptosis) and oncogenic miRNAs (activating oncogenesis or inhibiting the expression of suppressor genes) depending on their function. For example, miRNA-145, miRNA-34a, miRNA-29b, Let-7a, miRNA-340, miRNA-495, and miRNA-892b suppress the translation of oncogenes, consequently decreasing tumor growth, while miRNA-155 and miRNA-21 promote tumorigenesis by repression of tumor suppressor genes. Therefore miRNA-based therapies are an opportunity to restore the normal expression of miRNAs and the genes regulated by them in cancer. Since the first discovery of miRNAs, two therapeutic strategies have been developed. The first is based on the use of replacement therapies (restoring expression of miRNAs which have inhibited expression in cancer cells). The second is related to the inhibition of oncogenic miRNAs. Moreover, miRNA can modulate the TME of cancer [153,154,155]. In 2012, Mitra et al. reported that miRNA-155, miRNA-214, and mRNA-31 take part in the transformation of normal fibroblasts to CAFs in ovarian cancer. miRNA-155 was upregulated and miRNA-214 and miRNA-31 were downregulated in CAFs and CAF-stimulated samples. Transfection of human-derived normal omental fibroblasts with selected miRNAs promoted cell invasiveness, migration, and colony formation. In addition, CAFs could secrete CCL-5, which promotes the invasiveness of ovarian cancer cells [81]. In 2019, researchers reported that miRNA-222 expression was upregulated in CAFs compared to normal fibroblasts. Furthermore, transfection of normal fibroblasts with miRNA-222 led to their transformation to CAFs, while transfection of CAFs with miRNA-222 inhibitors suppressed CAF markers. Moreover, transfection of normal fibroblasts with miRNA-222 led to increased migration and invasion, whereas when CAFs were transfected with miRNA-222 inhibitors, these processes were downregulated. Interestingly, the miRNA-222 mimic and miRNA-222 inhibitors did not affect the proliferation of normal fibroblasts and CAFs, respectively [156].

In addition, miRNAs can affect immune cells in the TME [157]. In 2018, it was reported that miRNA-100 had the highest expression of selected miRNAs in the co-culture of macrophages and breast cancer cells. Moreover, miRNA-100 induced the polarization of macrophages from M1 to M2 and inhibited mTOR expression, and this inhibition led to the activation of the Hedgehog pathway in a syngeneic mouse breast cancer xenograft. This activation promoted the secretion of Interleukin-1 Receptor Antagonist (IL-1Ra), which is associated with the resistance of breast cancers to cisplatin. Of note, miRNA-100 inhibitors may suppress tumor progression [158]. Therefore, miRNA-based therapeutics represent great potential for future precision medicine cancer treatment.

#### 4.1.2. eRNA-Based Therapeutics for Cancer Treatment

MiRNAs can interact with loci-encoding eRNAs (enhancer RNAs) (Figure 2B). This may enhance the expression of neighboring genes by promoting a transcriptionally active form of chromatin. eRNA is a non-polyadenylated ncRNA transcribed from the DNA sequence. It interacts with the promoter of the targeted gene and activates the expression of the gene. eRNA may influence the process of gene expression by regulating the access of transcription factors to chromatin [159,160]. In the progression of metastatic breast cancer, a high risk of bone metastatic alterations is commonly observed. A recent screening of breast cancer and bone metastasis samples showed that one hundred and ninety-eight eRNA molecules have lower expression and eighty-four eRNA molecules have increased expression in breast cancer compared to normal tissue. Further research revealed that as the size of the tumor increases (to T3/T4 categories), the expression of eRNA is notably decreased compared to smaller tumors (T1, T2). eRNAs also affect immune cells in breast cancer tumors. The eRNA SLIT2 was reported to be downregulated in breast cancer cells compared to normal cells and associated with poor overall survival of breast cancer patients. The in vitro studies showed that inhibition of SLIT2 induced proliferation, migration, and invasion of MCF-7 cells in comparison to MCF-7 with overexpressed SLIT2. The proposed mechanism was that SLIT2 regulates the MAPK/c-FOS pathway, which affects cell survival [161]. Other studies confirmed that eRNAs are essential for the development of ovarian cancer. In 2022, it was reported that the eRNA ADCY10P1 as well as its target gene *NFYA* have reduced expression in ovarian cancer samples. In addition, three ovarian cancer cell lines (A2780, SKOV3, and OVCAR3) showed lower expression of ADCY10P1 than normal ovarian cell lines. It was presented that ovarian cancer cell lines overexpressing ADCY10P1 via a lentivirus vector had lower cell proliferation in comparison to cancer cell lines that were not transfected. Further studies confirmed that overexpression of ADCY10P1 in cancer cells inhibits migration and reduces the expression of proteins involved in metastasis (E-cadherin, N-cadherin, MMP-9). Finally, it was found that overexpression of this eRNA may be a positive prognostic factor in several cancers including bladder carcinoma, leukemia, and thymic carcinoma [162].

#### 4.1.3. CircRNA-Based Therapeutics for Cancer Treatment

Another important element regulating the transcription process is circRNA (circular RNA) (Figure 2C). circRNAs can consist of only introns, only exons, or introns and exon structures. They can modify the action of miRNAs by acting as a “miRNA sponge”. By blocking miRNA, circRNA can also stimulate gene expression. The intensive development of sequencing technology has allowed the identification of several circRNA molecules that are associated with the progression of cancer [163]. Fluorescence in situ hybridization analysis has shown decreased expression of the circRNA cirCCDC85A in breast cancer patients compared with healthy women. Similarly, in vitro studies demonstrated a negative relationship between circCCDC85A and processes involved in the proliferation, migration, and invasion of breast cancer cells. Moreover, circCCDC85A binds miRNA-550a-5p, which results in increased expression of MOB1A, linked with the Hippo pathway. These results show that increasing circCCDC85A in breast cancer may be a worthwhile aim of targeted treatment [164]. In 2021, it was demonstrated that elevated expression of miRNA-378 in ovarian cancer increases the expression of genes linked with tumor progression. Bioinformatics analysis revealed that miRNA-378 expression is regulated by circ-LOPD2, as high circ-LOPD2 expression leads to reduced expression of miRNA-378. Hence, increasing circ-LOPD2 expression in ovarian cancer may reduce cancer progression [165].

#### 4.1.4. lncRNA-Based Therapeutics for Cancer Treatment

Tumor development also depends on the disruption of regulatory mechanisms at the level of long non-coding RNA or lncRNA (Figure 2D), among others. LncRNA changes the chromatin structure of target genes, affecting the transcription of these genes. LncRNAs are involved in controlling cell proliferation (i.e., EPEL), apoptosis (i.e., PANDAR), metastasis (i.e., MALAT1), invasion (i.e., PVT1), migration (i.e., PVT1), immune evasion (NKILA), and EMT (TC0101441) [166,167,168,169,170]. In 2021, a comprehensive analysis of lncRNAs of over 9500 tumor samples (from 30 solid tumors) identified about 36 lncRNAs that are involved in the tumor immune microenvironment: ALOX12-AS1, AC100810.1, OSER1-DT, TRAM2-AS1, AP003486.1, SRP14-AS1, LINC00663, SLC25A5-AS1, AC091825.3, KIF9-AS1, NNT-AS1, Z99129.4, BCDIN3D-AS1, AC009163.7, AL158212.3, AL122035.1, TTLL1-AS1, AL049796.1, AC002398.1, AC007342.7, MAGI2-AS3, OXCT1-AS1, AL355297.2, AF287957.1, AC116535.2, Z97989.1, AL049838.1, AC004877.1, MIR497HG, PTPRD-AS1, ARHGEF26-AS1, AL157700.1, AC004947.2, AL109741.1, AC013553.3, and AL135960.1 [171]. In 2020, an analysis of hepatocellular carcinoma samples revealed the overexpression of NNT-AS1 in 62.5% of samples. Further research confirmed that NNT-AS1 may induce cancer progression through the promotion of EMT and induction of an immunosuppressive TME. In addition, NNT-AS1 activates the TGF-β pathway, which downregulates the immune response to cancer [172].

Recent data indicate that the lncRNA TCL6 is a marker of overall survival in renal cancer. Therefore, in 2020, Zhang et al. evaluated the role of TCL6 in tumor immune cells infiltrating breast cancer using TIMER (Tumor Immune Estimation Resource), which focuses on immune cells based on gene expression. The expression of TCL6 was linked with the activation of immune-related pathways including IL-6/JAK/STAT3. In addition, high expression of TCL6 was linked with high expression of PD-1 and CTLA-4 on T cells. The expression of TCL6 was also greater in ER^−^ and PR^−^ breast cancer cases [173].

The lncRNA group includes promoter-associated RNA (paRNA) [174] (Figure 2E). The mechanism of action of paRNA may be associated with the formation of complexes that affect epigenetic modulation. It has been demonstrated that changes in paRNA occur in cancers. Therefore, paRNA presents an interesting target for chemotherapy [175]. In 2019, researchers evaluated the effect of paRNA on CCND1 in Ewing sarcoma. CCND1 encodes cyclin D1, which is linked with tumor progression. Patients with Ewing sarcoma had notably higher expression of CCND1 in comparison with patients with other sarcomas and had higher levels of the paRNA pncCCND1-B. The knockdown of pncCCND1-B in TC71 cells resulted in the overexpression of CCND1 [176]. In 2020, Chen et al. studied CCDND1 amplification in over 6500 tumor tissue samples. The Cancer Genome Atlas database analysis showed that CCND1 amplification was linked with increased expression of CCND1 in esophageal cancer, head and neck squamous cell cancer, breast cancer, lung squamous cell cancer, bladder urothelial cancer, cholangiocarcinoma, hepatocellular cancer, cutaneous melanoma, stomach adenocarcinoma, and ovarian serous cystadenocarcinoma. Moreover, low CCND1 amplification in cancer samples was linked with an optimistic prognosis for anti–CTLA-4 therapy. This may be because elevated amplification of CCND1 is linked with a decrease in CD8^+^ T cells, cytotoxic T cells, DCs, and B cells and an increase in T regulatory cells in the TME [177].

#### 4.1.5. siRNA-Based Therapeutics for Cancer Treatment

Many scientists have tried to use siRNA in anticancer gene therapy (Figure 2F). The mechanism of action of siRNA is related to the formation of the RNA-induced silencing complex, which leads to gene silencing [178]. In 2016, researchers presented a nano-hydrogel system that allowed the efficient delivery of siRNA to ovarian cancer cells and silenced EGFR expression. Nanogels were chosen because of their high load capacity and porosity. Moreover, this structure allows modification of the surface for its functionalization [179]. Blackburn et al. used poly-N-isopropylmethacrylamide to encapsulate siRNA. The surface of the nanogel was functionalized by ε-maleimidocaproic acid. The developed system in combination with cisplatin therapy led to significant inhibition of tumor growth [180]. In 2022, researchers presented stealth magnetic siRNA nanovectors targeting survivin. The designed system decreased survivin expression in the HER2^+^ cell line SK-BR-3. In addition, preincubation of cells with the system decreased breast cancer cell survival [181].

#### 4.1.6. piRNA-Based Therapeutics for Cancer Treatment

piRNA (PIWI-interacting RNA) is involved in epigenetic and post-transcriptional modifications through the piRNA/PIWI complex proteins (Figure 2G). Unlike other ncRNAs, piRNAs are formed without the involvement of DICER [182,183]. The biogenesis of piRNA is not fully understood. It is assumed that it can be formed from lncRNAs, mRNAs, and transposons. The vast majority of piRNAs isolated to date correspond to retrotransposons. Activation of transposons can affect the integrity of the genome, which can lead to DNA damage. piRNA binds to PIWI proteins to create a piRNA-induced silencing complex, which can silence genes at the transcriptional and post-transcriptional levels [182]. In 2019, it was shown that piRNA-36712 may exhibit tumor suppressor properties in breast cancer development. Preliminary studies showed that breast cancer has low expression of this ncRNA compared to normal cells. In vitro studies have shown that increasing piRNA-36712 expression in MCF-7 cells decreases their proliferation, migration, and invasion. The anticancer activity of piRNA-36712 is linked with the regulation of SEPW1P and SEPW1. Increased expression of piRNA-36712 represses SEPW1 expression, which consequently increases p53 and p21 protein expression and affects the cell cycle and proliferation of breast cancer cells. In vivo studies have shown that MCF-7 xenografts with silenced piRNA-36712 expression had a 40% increase in the rate of lung metastasis compared to the scramble control, whereas, in xenografts with increased piRNA-36712 expression, tumor growth was inhibited. In contrast, in breast cancer cell lines carrying the mutated p53 variant (T47D, BT-474, MDA-MB-231), altering piRNA-36712 expression did not affect proliferation and metastasis. This indicates the importance of early genomic evaluation of the tumor before the acquisition of p53 variants to select the appropriate type of treatment [184].

#### 4.1.7. saRNA Small Activating RNA-Based Therapeutics for Cancer Treatment

saRNA is a double-strand RNA composed of 21 nucleotides. The chemical composition is similar to siRNA, but this system has the opposite function. saRNA in the cell combines with AGO2 and hnRNP (heterogeneous nuclear ribonucleoprotein). In this process, the passenger strand is discarded. Then, the created complex of saRNA-AGO2-hnRNP is transported to the nucleus where it binds to the promoter sequence of the targeted gene and promotes recruitment of RHA (RNA helicase A), RNAP II (RNA polymerase II), and CTR9 (a component of PAF complex) to DNA. This leads to transcriptional activation and synthesis of the targeted protein (Figure 2H) [185,186]. Therefore, saRNAs may be a promising therapeutic to increase the expression of downregulated genes in cancer. saRNAs may be used in combination therapy with other treatments or alone. In 2022, Lee et al. designed a saRNA-MAS1 system to increase *MAS1* expression in ovarian cancer, breast cancer, and pancreatic cancer. As a delivery system, researchers used amphiphilic dendrimers. It has been demonstrated that this technology leads to the activation of MAS1 expression in SKOV-3 and OVCA429 (ovarian cancer cells); HCC1937 and MDA-MB-231 (breast cancer cells); and AsPC1, CFPAC1, and PANC1 (pancreatic cancer cells). The delivery system used in the research increased the resistance of saRNA to RNase degradation compared to naked saRNA and provided high and effective transport of saRNA to cells. Moreover, in vivo studies confirmed that the designed saRNA system notably suppresses tumor growth in ovarian cancer and breast cancer xenografts. These results represent an innovative approach for increasing the expression of the targeted gene using amphiphilic dendrimer packed with saRNA [187].

Current clinical trials investigating the application of ncRNA in cancer treatment are summarized in Table 2.

### 4.2. Therapeutics Targeting ncRNAs in Cancer Treatment

Therapeutic approaches targeting ncRNAs include oligonucleotide therapeutics (antisense oligonucleotides (ASO), siRNAs, miRNAs, shRNA, gapmers, DNAzymes), antagomirs, small molecules, ncRNAs (siRNA, shRNA, or miRNA), and CRISPR/Cas9 systems [141]. Antisense oligonucleotides are short fragments of nucleic acids binding to RNA or DNA by hybridization. This inhibits the expression of the targeted gene. Antagomirs (anti-miRNA, blockmirs) are RNA analogs with various modifications (2′-O–ME of ribose, phosphorothioate bond, and cholesterol structure at 3′-end). The sequence of antagomir is complementary to the targeted miRNA sequence [215]. The next essential group of therapeutics targeting ncRNAs in the TME is small molecules. Design strategies for these drugs include a high-throughput screening approach (based on the combinatorial chemistry), a small-molecule microarray approach (based on binding of small molecules to motif in ncRNA), structure-design approach (based on modeling miRNA using MC-Fold for screening small molecules), phenotype screening approach (based on measurement of effect in the biological system), fragment-based approach (based on creating library of compounds consisting of fragments of molecules), pharmacological validation approach (based on the literature and pharmacological experiments), one bead two compounds (based on combinatorial library), and others [216,217].

ncRNA in the TME may be targeted also by another ncRNA. In 2019, Lu et al. presented prominent results of silencing lncRNA DANCR (significantly overexpressed in triple-negative breast cancer) using siRNA packed in an RGD-PEG-ECO (arginine/glycine/aspartate peptide—polyethylene glycol -1-aminoethyliminobis[N-oleicylcysteinyl-1-aminoethylpropionamide) nanoparticle. The designed system notably suppressed invasion and proliferation of MDA-MB-231 and BT549 breast cancer cell lines via pathways linked with B-catenin, ZEB, N-Cadherin, and EXH2. Moreover, in in vivo studies, RGD-PEG-ECO nanoparticles inhibited and reduced tumor volumes compared to untreated mice without affecting the mice’s body weight. This shows that ncRNAs can be targeted also using other ncRNAs [218].

Another group of compounds that may have potential applications in targeted therapy against ncRNAs in TME is gapmers. Gapmers are chemically modified oligonucleotides that can activate RNase H, which cuts the mRNA strand and inhibits translation [219]. The tremendous potential that this group of compounds holds is demonstrated in Table 3 [220,221,222,223,224,225,226,227,228,229,230,231,232,233,234,235,236,237,238,239,240,241,242,243,244,245,246,247,248,249,250,251].

One of the lncRNAs overexpressed in lung cancer is *MALAT1.* Overexpression of *MALAT1* is linked with increased tumor growth, colony formation, as well as invasiveness. In 2013, Gutchner et al. showed that *MALAT1* gapmer can significantly decrease *MALAT1* expression in vivo and in vitro. Moreover, the presented oligonucleotide inhibited tumor growth and the number of metastases in a mouse model. This shows that application of gapmers may be crucial to develop promising therapeutics in lung cancers [243].

Around 10% of skin cancers are associated with overexpression of MITF, whose expression is regulated by lncRNA SAMMSON. Therefore, Leucci et al. knocked down SAMMSON using a gapmer. The results showed that the knockdown of SAMMSON led to the activation of apoptosis via the intrinsic apoptotic pathway. The in vivo studies demonstrated that the gapmer for SAMMSON is a promising therapeutic to decrease tumor volume, inhibit proliferation, and activate apoptosis in tumor cells. This result shows the enormous potential of gapmers in the future anticancer therapy [251].

DNAzymes are engineered single-stranded catalytic oligonucleotides that bind to targeted RNA sequences. In consequence, it leads to the cleavage of RNA. Most of the studies based on DNAzymes are focused on targeting mRNA. However, these systems may also be used in the inhibition of miRNA expression. In 2019, Veedu et al. designed a DNAzyme against miRNA-21 (RNV541). The designed system led to significant inhibition of miRNA-21 expression in glioblastoma cell line U87MG and MDA-MB-231 breast cancer cells. This shows the great potential DNAzymes have to design novel effective anticancer therapeutics [253].

One of the latest methods to modify ncRNAs’ expression in the TME is CRISPR technology. In 2021, Kimura et al. using Cas9 nickase (Cas9n) combined with sgRNA inhibited the expression of miRNA-146b in an aggressive anaplastic thyroid cancer cell line (KTC2), which also decreased cell growth, migration, and invasion. Researchers also showed the prominent effect of the Cas9n system on the inhibition of tumor growth in vivo. These findings are the first approach to treat thyroid cancer using the CRISPR system to target miRNA [254].

The promising features of compounds targeting non-coding RNAs have resulted in several clinical trials utilizing them. Current clinical trials are summarized in Table 4.

## 5. Achievements and Future Perspectives

The pioneers of the study of TME are considered to be Virchow and Paget [270,271], who were the first to recognize the connection between the tumor and the inflammatory process [270,271]. In the first stages of cancer, the microenvironment inhibits tumor growth, but as the disease progresses, tumor cells reprogram the TME, creating favorable conditions that stimulate tumor growth and proliferation. Involved in the regulation of the TME are various cytokines and chemokines. Hence, a thorough understanding of the interactions between the tumor and its surrounding environment could provide potential targets for new anticancer therapies [270,271]. For this purpose, modern bioinformatics solutions are being used to analyze gene expression, identify pathways affecting tumorigenesis, and identify cells included in the TME. A technology that enables cell-to-cell analysis of the TME is RNA sequencing. This method has the advantage of being compatible with the preserved material (frozen or formalin-fixed and paraffin-embedded), minimizing the time needed to prepare libraries and reducing the cost of performing the analysis. The collected data of bioinformatics analysis allowed for the identification of ncRNAs, which are important in cancer transformation [9,10,12,13,14,15,16].

Nowadays, artificial intelligence (AI) is gaining importance in medicine. AI as applied in cancer research is a combination of algorithms that help clinicians to analyze multi-omic (genomic, proteomic, transcriptomic, epigenetic, histochemistry, MRI, X-rays, etc.) data from cancer tissues from all over the world. This approach provides information about genes that are mutated in various cancers and which genes are prominent in the development of cancer. This information may also be utilized in the early diagnosis of cancer. AI methods are capable of not only analyzing the data but also learning new information based on the results. A bioinformatics approach presented in 2021 demonstrated how the usage of AI may be used to harness the interactions between cancer and the TME and improve the survival rates of breast cancer patients. In the study, four methods were used to analyze TME immune cells according to patient survival rate: EPIC, CIBERSORT, TIMER-, and xCell explainable artificial intelligence models [100]. The comprehensive analysis revealed that patients with a high proportion of T cells, B cells, and NK cells had a higher rate of survival compared to patients with lower proportions of these immune cells. According to the TIMER and xCell bioinformatics results, T cells were key for patient survival. On the other hand, according to the EPIC analysis, B cells had a prominent function in a high rate of overall survival. This shows how various analyses of a dataset using different algorithms reveal new links between immune cells, cancer cells, and survival rates. In the future, interpretation of the TME profile may aid early diagnosis of cancer and evaluation of response to chemotherapy [100]. In addition, researchers are working intensively on using AI in cancer image analysis. The development of appropriate algorithms that recognize specific alterations in magnetic resonance imaging or computed tomography scans will be a breakthrough in cancer diagnostics. The most difficult challenge in AI analysis of imaging will be to create a sufficiently large database that makes it possible to distinguish healthy and cancerous cells [272].

Alterations in the expression profile of non-coding RNA can mediate the relationship between cancer cells and the TME, becoming instrumental in cancer development and progression. A variety of approaches utilizing ncRNA to create personalized therapies and diagnostic methods are possible and underway.

## Figures and Tables

**Figure 2 ncrna-09-00052-f002:**
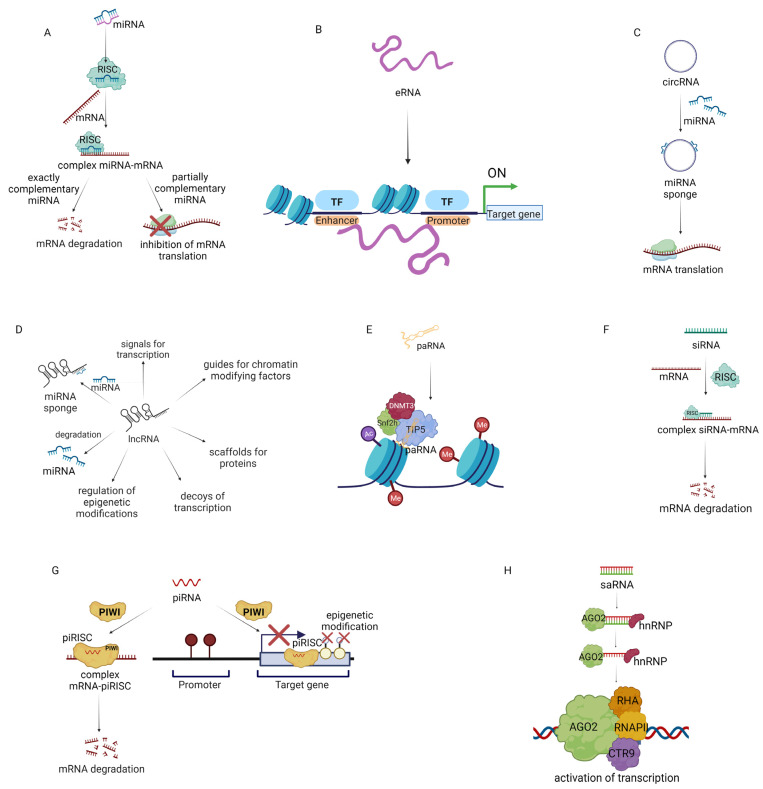
Mechanism of action of non-coding RNA (ncRNA). (**A**) miRNA inhibits gene expression by degradation of mRNA or inhibition of translation. (**B**) eRNA (enhancer RNA) promotes gene expression. (**C**) circRNA (circular RNA) acts as a sponge of miRNA. (**D**) Multi-directional influence of lncRNA (long non-coding RNA) on gene expression. (**E**) paRNA (promoter-associated RNA) inhibits gene expression. (**F**) siRNA (small interfering RNA) inhibits gene expression. (**G**) piRNA (PIWI-inhibiting RNA) inhibits gene expression by mRNA degradation and epigenetic modulations. (**H**) saRNA (small activating RNA) activates the expression of the targeted gene [148,149,150,151,152].

**Table 1 ncrna-09-00052-t001:** Methods of transcriptome analysis. Abbreviations: s4U, 4-thiouridine; EU, 5-Ethynyluridinee; Bru, 5′-bromouridine; RT-qPCR, reverse transcription–quantitative polymerase chain reaction.

Method	Brief Description	Advantages	Disadvantages	References
RT-qPCR	Technique based on PCR	High sensitivity	Time-consuming; requires primers; Allows for the evaluation of the expression of known transcripts	[9]
Microarray	Technique based on the ability of complementary nucleic acid molecules to form double-stranded structures	Well-defined protocols for hybridization	Stringent criteria for sample collection requires a high quantity of RNA; expensive	[10]
Bulk RNA-seq	Technique to evaluate mean gene expression of thousands of cells	Costs less than single-cell and spatial RNA-seq; requires less time compared to single-cell RNA-seq and spatial RNA-seq	Less detailed information on individual cells provide averaged gene expression in whole cells	[11]
Single-cell RNA-seq	Sequencing technique to evaluate mRNA in a single cell	Enables analysis of the whole transcriptome including non-coding sequences; low background noise	Requires adequate preparation of tissue/cells; expensive; time-consuming; does not provide spatial information on transcriptome	[12]
Spatial RNA-seq	Sequencing technique for evaluation of mRNA in the tissue area	Analyzes the whole transcriptome evaluates the interactions between cancer cells and tumor microenvironment	Requires adequate preparation of tissue/cells; cannot analyze a single cell; expensive; time-consuming	[13]
Metabolic labeling	Technique in which RNA is labeled with uracil analogs (s4U, EU, Bru) in cell culture	Efficient; well-validated; high resolution	Uracil analogs are cytotoxic	[14]
Nuclear run-on	Technique in which RNA in the isolated nucleus is labeled	Measurement of transcription in the primary state; allows distinct transcription and post-transcription changes in gene expression	Requires the ice-cold temperature to isolate nuclei; requires a large number of cells; results are dependent on the efficient induction of transcription outside the cell	[15]
Chromatin immunoprecipitation (ChIP)	Technology based on antibodies to selectively isolate DNA-binding proteins and their DNA targets	Allows monitoring of changes in a single promoter in a time-dependent manner; may be used to follow transcription factors in the whole human genome	Low resolution; expensive; risk of protein rearrangement during analysis	[16]

**Table 2 ncrna-09-00052-t002:** List of clinical trials associated with non-coding RNA used in cancer treatment.

Type of ncRNA	Clinical Trial	Objectives of Study	Cancer	Reference
EphA-2 siRNA (siRNA)	Phase 1 (recruiting)	Silencing EphA-2	Solid tumors	[188,189]
CpG-STAT3 siRNA CAS3/SS3 (CpG oligonucleotide and siRNA)	Phase 1 (recruiting)	Downregulation of STAT3	Relapsed B-cell non-Hodgkin lymphomas	[190]
KRAS G12- LODER (siRNA)	Phase 1 (recruiting)	Silencing KRAS G12D	Pancreatic cancer	[191,192]
	Phase 2 (recruiting)	Silencing KRAS G12D to increase the anticancer activity of gemcitabine in combination with nab-paclitaxel or FOLFIRINOX chemotherapy	Pancreatic cancer	[193]
GSTP siRNA/NBF-006(siRNA)	Phase 1 (recruiting)	Silencing GSTP to decrease KRAS signaling pathway	NSCLC Pancreatic cancer Colorectal cancer	[194,195]
Two siRNAs: TGF-β1 COX-2 /STP705 (siRNA)	Phase 2 (not recruiting)	Silencing expression of TGF-β1 and COX-2 to inhibit cell survival and induce tumor cell apoptosis	Cutaneous squamous cell carcinoma skin cancer	[196,197]
DCR-MYC (siRNA)	Phase 1/2 (terminated)	Inhibition of MYC expression	Hepatocellular carcinoma	[198]
	Phase 1 (terminated)	Inhibition of MYC expression	Solid tumors Multiple Myeloma Non-Hodgkins Lymphoma Pancreatic Neuroendocrine Tumors Primary Central Nervous system tumors (PNET) N	[199]
TBI-1301 (siRNA)	Phase 1 (active)	Silencing endogenous TCR on T cells in combination with cyclophosphamide and fludarabine	Synovial Sarcoma Melanoma Esophageal Cancer Ovarian Cancer Lung Cancer Bladder Cancer Liver Cancer	[200]
	Phase 1/2 (active)	Combination of TBI-1301 and cyclophosphamide	Synovial Sarcoma	[201]
	Phase 1 (unknown)	Combination of TBI-1301, cyclophosphamide and fludarabine	Solid tumors	[202]
TKM-080301 (siRNA)	Phase 1/2 (completed)	Inhibition of PLK1 expression	Hepatocellular carcinoma Hepatoma Liver cancer	[203]
	Phase 1/2 (completed)		Neuroendocrine Tumors Adrenocortical Carcinoma	[204]
	Phase 1 (completed)		Solid cancers with hepatic metastases	[205]
INT1-B3 (mimic miRNA)	Phase 1 (recruiting)	Mimic miRNA-193a-3p	Solid tumors	[206]
MRX34 (mimic miRNA)	Phase 1/2 (withdrawn)	Combination of MRX34 and dexamethasone	Melanoma	[207]
	Phase 1 (terminated)	Mimic miRNA-34	Primary Liver Cancer SCLC Lymphoma Melanoma Multiple Myeloma Renal Cell Carcinoma NSCLC	[208]
Mesomir-1 (mimic miRNA)	Phase 1 (completed)	Mimic miRNA-16	Malignant Pleural Mesothelioma NSCLC	[209]
MTL-CEBPA (saRNA)	Phase 2 (recruiting)	Increase C/EBP-α expression	Hepatocellular carcinoma	[210,211]
	Phase 1 (active)	Combination therapy of MTL-CEBPA and pembrolizumab	Breast cancer Lung Cancer Ovarian Cancer Pancreatic cancer Gall bladder cancer Hepatocellular cancer Neuroendocrine cancer Cholangiocarcinoma	[212]
	Phase 1 (active)	Combination therapy MTL-CEBPA and sorafenib	Hepatocellular carcinoma	[213]
	Phase 1 (recruiting)	Combination therapy MTL-CEBPA and atezolizumab and bevacizumab	Hepatocellular carcinoma	[214]

**Table 3 ncrna-09-00052-t003:** Selected Gapmers with anticancer activity.

Gapmer	Cancer	Function	Type of Research	Reference
CT102 gapmer for IGF1R mRNA	HCC	Inhibition of PI3K/AKT pathway, Induction of apoptosis in tumor cells by interaction with GAS2, POLA2, LGALS2	In vitro, in vivo	[220]
Gapmers for ALKBH5 or FTO	Clear Renal Cell Carcinoma	Inhibition of migration, proliferation of tumor cells, downregulation of Vimentin and PCNA	In vitro	[221]
Gapmer for HIF1A-As2	NSCLC	Increase sensitivity of NSCLC tumors to MYC inhibitor (10058-F4) and cisplatin treatment Inhibition of colony formation, spheroid formation in vitro Inhibition of tumor growth in PDX (patient-derived xenograft) model	In vitro, in vivo	[222]
Gapmer for *SRRM4*	NSCLC, Prostate cancer	Reduction in cell growth	In vitro	[223]
Gapmer for SOX12	Human Acute Myeloid Leukemia Cells	Inhibition of expression of *SOX12* Activation of apoptosis by increased activity of caspase 3 and 9	In vitro	[224]
Gapmer for Smyca	Breast cancer	Inhibition of TGF-β/Smad and c-Myc pathways Inhibition of tumor growth	In vitro In vivo	[225]
Gapmer for GGCT	Lung cancer	Inhibition of expression of GGCT to decrease the viability of tumor cells Activation of apoptosis via caspase 3 and 8 Activation of AMPK Inhibition of tumor growth	In vitro, in vivo	[226]
Gapmer for p53 mutant protein	Breast and pancreas cancers	Inhibition of cell viability and proliferation Decrease expression of proapoptotic protein Bcl-2	In vitro	[227]
Gapmer for lncRNA MIR100HG	Acute Megakaryocyte Leukemia	Inhibition of lncRNA MIR100HG Induction of apoptosis Increased level of TGF-B expression	In vitro	[228]
G3139 for Bcl-2 mRNA	Breast cancer	Reduction in cell viability of breast cancer cells Induction of apoptosis Inhibition of tumor growth Decreasing expression of Bcl-2	In vitro, in vivo	[229]
Gapmer for SRRM4	SCLC	Inhibition of miRNA-4516 expression Inhibition of cell growth	In vitro, in vivo	[230]
Gapmer for NEAT1	Multiple Myeloma	Inhibition of cell Activation of caspase 3 Inhibition of cell proliferation Inhibition of tumor growth	In vitro, in vivo	[231]
Gapmer ISTH0047 and ISTH2047	Glioblastoma	Decrease expression of TGFB1/2 Inhibition of migration and invasion Increased survival of rodent glioma models in vivo	In vitro, in vivo	[232]
Gapmer for b2a2 and b3a2 *BCR/ABL*	Leukemia	Inhibit viability of cells Activation of executive caspases 3/7	In vitro	[233]
SPC3042	Prostate cancer	Increase expression of caspase 3/7 activity Decrease expression of antiapoptotic Bcl-2 mRNA Decrease tumor weight in combination with taxol	In vitro, in vivo	[234]
Gapmer for DNp73	Lung cancer, Melanoma	Induction of apoptosis Inhibition of tumor growth	In vitro, in vivo	[235]
Gapmer for *let-7* miRNA	Multiple myeloma	Decrease expression of MYC, KRAAS, CCND1, E2F6, DICER1, HMGA1 Inhibition of tumor growth	In vitro, in vivo	[236]
Gapmer for Bcl-2	Lung cancer	Induction of apoptosis Decrease expression of Bcl-2 mRNA Inhibition of tumor growth	In vitro, in vivo	[237]
Gapmer targeting TGF-B3	Glioblastoma	Decrease expression of pSMAD2, Inhibition of invasion, Inhibition of tumor growth	In vitro, in vivo	[238]
Gapmer for BC200	Breast cancer, Hepatocellular cancer, Lung cancer, Ovarian Cancer	Induction of apoptosis	In vitro	[239]
GapmeR for XLOC_109948	Acute Myeloid Leukemia	Induction of apoptosis	In vitro	[240]
Gapmer for lncRNA MALAT1	Multiple Myeloma	Decrease expression of MALAT1, inhibition of colony formation, inhibition of tumor growth	In vitro, in vivo	[241]
Gapmer for MALAT1	Multiple Myeloma	Increased activation of PAR signaling Induction of caspase-3 activity Activation of apoptosis	In vitro, in vivo	[242]
Gapmer for MALAT1	Lung cancer	Decreased cell migration, reduced number of nodules	In vitro, in vivo	[243]
Gapmer for MALAT1	Multiple Myeloma	Induction of apoptosis Inhibition of tumor growth	In vivo	[252]
Gapmer for miR-17–92s	Multiple Myeloma	Reduced viability of cells, induction of apoptosis	In vitro, in vivo	[244]
Gapmer for lncRNA PVT1	Acute Erythroleukemia	Induction of apoptosis Inhibition of c-MYC expression	In vitro	[245,246]
Gapmers for Bcl-XL	Lung cancer	Inhibition of Bcl-2 expression Induction of apoptosis by activation of caspase 3	In vitro	[247]
Gapmer for p21	Breast cancer	Inhibition of ERα expression	In vitro	[248]
Gapmer for Bcl-2	Breast cancer	Inhibition of cell viability Activation of caspase 3	In vitro	[249]
Gapmer for Clusterin	Breast cancer	Induction of apoptosis in combination with trastuzumab	In vitro	[250]
Gapmer for SAMMSON	Melanoma	Inhibition of lncRNA *SAMMSON*	In vitro, in vivo	[251]

**Table 4 ncrna-09-00052-t004:** List of therapeutics targeting ncRNAs in clinical trials.

Name of Drug (Type of Drug)	Clinical Trial Phase	Objectives of Study	Cancer	Reference
Anti-mir-10b (antagomir)	Diagnostic (Recruiting)	Testing in vitro sensitivity of individual primary tumors	Glioblastomas	[255]
Cobomarsen/MRG-106 (antagomir)	Phase 2 (Terminated)	Inhibition of miRNA-155	Cutaneous T-Cell Lymphoma	[256]
	Phase 2 (Terminated)	Combination of cobomarsen with vorinostat	Cutaneous T-Cell Lymphoma	[257]
	Phase 1 (Completed)	Inhibition of miRNA-155	Cutaneous T-cell Lymphoma Chronic Lymphocytic Leukemia Diffuse Large B-Cell Lymphoma Adult T-cell Leukemia	[258]
OGX-011	Phase 1 (Completed)	Increase anticancer activity of flutamide and buserelin	Prostate cancer	[259]
	Phase 1 (Completed)	Combination therapy with docetaxel	Solid tumors	[260]
	Phase 2 (Completed)	Combination therapy with docetaxel	Breast cancer	[261]
OGX427 (Apatorsen)	Phase 1 (Unknown)	Inhibition of Hsp27 to treat patients with advanced cancers	Advanced cancer	[262]
	Phase 1 (Completed)	Safety of OGX427 in the treatment of prostate cancer, ovarian cancer, NSCLC, breast cancer, bladder cancer	Cancers	[263]
	Phase 2 (Completed)	Inhibition of Hsp27 to slow the progression of prostate cancer	Prostate cancer	[264]
	Phase 2 (Completed)	Combination therapy of OGX-427 and docetaxel	Bladder carcinoma	[265]
Danvatirsen (AZD9150)	Phase 2 (Recruiting)	Effectiveness of combination therapy of danvatirsen and pembrolizumab	HNSCC	[266]
	Phase 2 (Active, not recruiting)	Effectiveness of combination therapy davatirsen and durvalumab	Advanced and refractory pancreatic, non-small cell lung cancer, colorectal cancer	[267]
	Phase 2 (Completed)	Effectiveness of combination therapy of duralumab with oleclumab or monalizumab or danvatirsen	Lung cancer	[268]
	Phase 1 (Active, not recruiting)	Effectiveness of durvalumab with davatirsen/oleclumab/carboplatin/gemcitabine/cisplatin/Nab-paclitaxel	NSCLC	[269]

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
