# Peer review of "Non-Coding RNAs: Foes or Friends for Targeting Tumor Microenvironment"

_ncrna, 2023, doi:10.3390/ncrna9050052_

Round 1

Reviewer 1 Report

This is a very comprehensive treatment of the topic of ncRNAs in cancer therapy. There is definitely a need for this work to be published but some more work to integrate the cancer sections with discussion of the non-coding RNA knowledge and potential targeting would help to raise its significance. The manuscript is also very ambitious and could benefit by giving additional attention to the literature. For instance, the topic of exosomes, non-coding RNAs, and cancer has been the subject of many review articles over the last five years, but this article does not mention many of these: e.g.,

https://www.nature.com/articles/s41388-019-1040-y

https://www.cell.com/molecular-therapy-family/nucleic-acids/fulltext/S2162-2531(20)30310-3

https://www.cell.com/molecular-therapy-family/nucleic-acids/fulltext/S2162-2531(22)00147-0

https://link.springer.com/article/10.1186/s12943-019-0984-4#Bib1

https://www.nature.com/articles/s41392-022-00975-3)

Minor comments:

Minor typos and issues noted in the attached marked up PDF.

Author Response

Reviewer #1

Query 1: This is a very comprehensive treatment of the topic of ncRNAs in cancer therapy. There is definitely a need for this work to be published but some more work to integrate the cancer sections with discussion of the non-coding RNA knowledge and potential targeting would help to raise its significance. The manuscript is also very ambitious and could benefit by giving additional attention to the literature. For instance, the topic of exosomes, non-coding RNAs, and cancer has been the subject of many review articles over the last five years, but this article does not mention many of these: e.g.,

https://www.nature.com/articles/s41388-019-1040-y

https://www.cell.com/molecular-therapy-family/nucleic-acids/fulltext/S2162-2531(20)30310-3

https://www.cell.com/molecular-therapy-family/nucleic-acids/fulltext/S2162-2531(22)00147-0

https://link.springer.com/article/10.1186/s12943-019-0984-4#Bib1

https://www.nature.com/articles/s41392-022-00975-3)

1) Response:

  • Thank you very much for the comprehensive review and positive comments. We appreciate for drawing our attention to the last review articles. We have added them to the presented review. https://www.nature.com/articles/s41388-019-1040-y: Please refer to lines 192-198; 204-209;
  • https://www.cell.com/molecular-therapy-family/nucleic-acids/fulltext/S2162-2531(20)30310-3: Please refer to lines: 192-201
  • https://www.cell.com/molecular-therapy-family/nucleic-acids/fulltext/S2162-2531(22)00147-0: Please refer to lines: 232-239
  • https://link.springer.com/article/10.1186/s12943-019-0984-4#Bib1: please refer to lines: 240-243
  • https://www.nature.com/articles/s41392-022-00975-3: please refer to lines 560-568, 811-814

    Query 2: Minor comments:

    Minor typos and issues noted in the attached marked up PDF.

    Line 101-107 was various format of lncRNA ZFAS1 . We unified formatting.

    2) Response:

    We thank the Reviewer for the suggestions. We revised the manuscript according to the reviewer’s suggestions:

    We formatted lncRNA ZFAS1 according to Reviewer’s suggestions.  (please refer to lines 120-127)

    • In line 124 “paly” was changed to “play” (please refer to line 143)
    • In line 153 “are small molecules: we have changed word molecules for vesicles (please refer to line 172)
    • In lines 160 to 161, Reviewer suggested joining the sentences, we have rewritten this paragraph (please refer to 176-246)
    • In line 164, we corrected the misspelling of nanocarriers  (please refer to line 179)
    • In line 212, “utilize” was corrected for “utilizes” (please refer to line 287)
    • In line 245,  “Tcells” was corrected with “T cells” (please refer to line 320)
    • In line 311, as suggested, we added the information regarding connecting RNA-sequencing with ncRNA in breast cancer. Please refer to 396-406.
    • In line 330, as suggested, we added the citation[109] Goff, S.L.; Danforth, D.N. The Role of Immune Cells in Breast Tissue and Immunotherapy for the Treatment of Breast Cancer. Clinical Breast Cancer 2021, 21, e63-e73, doi:https://doi.org/10.1016/j.clbc.2020.06.011.(please refer to line 408)
    • In line 336, as suggested, we added the information regarding connecting RNA-sequencing with ncRNA in lung cancer. (please refer to lines 415-421)
    • In line 379, as suggested, we added the discussion about ncRNA, (please refer to lines 452-456)
    • In line 427, the misspelling “fetal” was changed with fatal (please refer to line 499)

    - In paragraph 4,   we rewrite this part. According to the suggestions, we have improved the paragraph (please refer to lines 560-568)

    - In line 696, we reworded the paragraph. Please refer to lines 587-594

    - In line 711, we added the citation Virchow.(please refer to line 871, citation 222; line 1445)

    - In line 734, we added the citation [96] Chakraborty, D.; Ivan, C.; Amero, P.; Khan, M.; Rodriguez-Aguayo, C.; Başağaoğlu, H.; Lopez-Berestein, G. Explainable Artificial Intelligence Reveals Novel Insight into Tumor Microenvironment Conditions Linked with Better Prognosis in Patients with Breast Cancer. Cancers (Basel) 2021, 13, doi:10.3390/cancers13143450. Please refer to line 897.

    - The paragraph in lines 750-755 was removed due to the fact it was not connected with ncRNA

Reviewer 2 Report

Szymanowska and colleagues present a useful comprehensive account of present knowledge of ncRNAs in TME and potential therapeutic/diagnostic opportunities. The review is primarily focussed on summarising literature in various areas. It would benefit from some more effort to synthesise, interpret and look to future challenges. Some areas of the paper are hard to follow and assume a rather nonstandard view of the field, which will confuse many readers in its present form.

The paper has an important focus, yet this is not mentioned in the title. I find the title does not accurately reflect the content.

Much space is devoted to RNA-sequencing in various cancer types. Yet little is devoted to the caveats and drawbacks of RNA-seq analyses. For example:

- is differential expression a good proxy for functional disease genes?

- what should control sample should tumour expression be compared to, to find differentially expressed genes? 

- how much heterogeneity at individual cell, and cell subtypes, is overlooked in bulk RNAseq?

Mutation: There is no mention of how germline and somatic mutations impact ncRNAs. 

Therapeutics: This section is rather superficial. The principal therapeutic strategies, eg oligonucleotide vs small molecule vs ncRNA fragments, are not mentioned. Nor is the critical hurdle of oligonucleotide delivery. 

Section 4.7: There is text in this section is rather confusing. Only after quite some reading did this Reviewer realise the authors are referring to therapies based on ncRNA delivery itself, rather than inhibition of ncRNAs. This should be rewritten to clarify that point. The standard of ncRNA therapy is inhibition with Gapmer ASOs or other strategies. These are not mentioned anywhere in the paper, but rather the authors are assuming a rather nonstandard methodology of ncRNA direct delivery. I would suggest to consult colleagues with expertise in this topic to expand and clarify it. 

ok

Author Response

Reviewer #2:

Query 1: Szymanowska and colleagues present a useful comprehensive account of present knowledge of ncRNAs in TME and potential therapeutic/diagnostic opportunities. The review is primarily focussed on summarising literature in various areas. It would benefit from some more effort to synthesise, interpret and look to future challenges. Some areas of the paper are hard to follow and assume a rather nonstandard view of the field, which will confuse many readers in its present form.

The paper has an important focus, yet this is not mentioned in the title. I find the title does not accurately reflect the content.

1) Response:

We thank the Reviewer for revising our manuscript. We changed the title to “Non-coding-RNA a breakthrough in treatment and cancer analysis”.

 Query 2: Much space is devoted to RNA-sequencing in various cancer types. Yet little is devoted to the caveats and drawbacks of RNA-seq analyses. For example:

- is differential expression a good proxy for functional disease genes?

- what should control sample should tumour expression be compared to, to find differentially expressed genes?

- how much heterogeneity at individual cell, and cell subtypes, is overlooked in bulk RNAseq?

Mutation: There is no mention of how germline and somatic mutations impact ncRNAs.

2) Response:

We thank the Reviewer for the suggestions. We believe that our review is unique due to the potential use of t RNA sequencing to define the novel promising targets for the treatment of specific non-coding RNAs.

We added the missing information:

- Please refer to lines 384-388 for differential expression a good proxy for functional disease genes.

- Please refer to lines 385-388 for: what should control sample should tumor expression be compared to, to find differentially expressed genes?

- Please refer to lines 388-393 for: how much heterogeneity at the individual cell, and cell subtypes, is overlooked in bulk RNAseq?

Please refer to lines 81-99 for Mutation: There is no mention of how germline and somatic mutations impact ncRNAs.

Query 3: Therapeutics: This section is rather superficial. The principal therapeutic strategies, eg oligonucleotide vs small molecule vs ncRNA fragments, are not mentioned. Nor is the critical hurdle of oligonucleotide delivery.

3) Response:

We thank you the Reviewer for the comment. The article has been supplemented with the therapeutics strategies including oligonucleotides and small molecules.

Please refer to lines: 559-568, and lines 799-856

Query 4: Section 4.7: There is text in this section is rather confusing. Only after quite some reading did this Reviewer realize the authors are referring to therapies based on ncRNA delivery itself, rather than inhibition of ncRNAs. This should be rewritten to clarify that point. The standard of ncRNA therapy is inhibition with Gapmer ASOs or other strategies. These are not mentioned anywhere in the paper, but rather the authors are assuming a rather nonstandard methodology of ncRNA direct delivery. I would suggest to consult colleagues with expertise in this topic to expand and clarify it.

4) Response:

Thank you very much for your valuable suggestion for this section. We rewrote and changed paragraph 4. Please refer to lines:559-568, 769-793, 810-869

Round 2

Reviewer 2 Report

> I thank the Reviewers for considering my comments. However I think that my comments were not clear because they have not been fully responded to. New Reviewer comments are indicated by '>'. I invite the Reviewers to think a litttle bit more about these issues, since the present responses and changes are insufficient.

1) Response:

We thank the Reviewer for revising our manuscript. We changed the title to “Non-coding-RNA a breakthrough in treatment and cancer analysis”.

> Apologies. The point of my comment is that the Review seems to be dedicated to Tumor Microenvironment, yet this is not mentioned in the title. If you do not think its necessary, then please keep present title.

 Query 2: Much space is devoted to RNA-sequencing in various cancer types. Yet little is devoted to the caveats and drawbacks of RNA-seq analyses. For example:

- is differential expression a good proxy for functional disease genes?

- what should control sample should tumour expression be compared to, to find differentially expressed genes?

- how much heterogeneity at individual cell, and cell subtypes, is overlooked in bulk RNAseq?

Mutation: There is no mention of how germline and somatic mutations impact ncRNAs.

2) Response:

We thank the Reviewer for the suggestions. We believe that our review is unique due to the potential use of t RNA sequencing to define the novel promising targets for the treatment of specific non-coding RNAs.

We added the missing information:

- Please refer to lines 384-388 for differential expression a good proxy for functional disease genes.

> The point was, the assumption is frequently made that differential expression indicates functional disease genes. Yet the goodness of this assumption should be challenged more frequently. How many changing genes are actually causally implicated in disease and are valuable therapeutic targets? Are researchers wasting time and effort blindly following this assumption.

- Please refer to lines 385-388 for: what should control sample should tumor expression be compared to, to find differentially expressed genes?

> The point was, what normal cell sample should be used? If a tumor originates from a single stem cell in a complex tissue mixture comprising a heterogeneous mixture of cell types, then the bulk tumor sample (which originated from that stem cell) is likely to have very different gene expression characteristics from the surrounding tissue, independently of its tumor status. Researchers rarely consider this issue. It is somewhat unavoidable, but could perhaps be mitigated by single cell approaches.

- Please refer to lines 388-393 for: how much heterogeneity at the individual cell, and cell subtypes, is overlooked in bulk RNAseq?

> Bulk RNAseq averages across all cells, particularly overlooking rare but possibly important cell types (including originator cells of the tumour possibly).

Please refer to lines 81-99 for Mutation: There is no mention of how germline and somatic mutations impact ncRNAs.

> Again, quite a superficial response. How do those somatic mutations affect microRNAs to promote tumor growth? What about germline variants? There is a publication from Calin's group on a germline variant (Redis et al). 

Query 3: Therapeutics: This section is rather superficial. The principal therapeutic strategies, eg oligonucleotide vs small molecule vs ncRNA fragments, are not mentioned. Nor is the critical hurdle of oligonucleotide delivery.

3) Response:

We thank you the Reviewer for the comment. The article has been supplemented with the therapeutics strategies including oligonucleotides and small molecules.

Please refer to lines: 559-568, and lines 799-856

> Thank you for making these changes. However, in the section on Gapmers, the example provided concerns a protein coding gene (GGCT). Since there are so many studies targeting ncRNAs in cancer using Gapmers, I am puzzled why a protein coding example was chosen. Nice cases to use could be MALAT1 (Gutschner) or SAMMSON (Leucci).

Query 4: Section 4.7: There is text in this section is rather confusing. Only after quite some reading did this Reviewer realize the authors are referring to therapies based on ncRNA delivery itself, rather than inhibition of ncRNAs. This should be rewritten to clarify that point. The standard of ncRNA therapy is inhibition with Gapmer ASOs or other strategies. These are not mentioned anywhere in the paper, but rather the authors are assuming a rather nonstandard methodology of ncRNA direct delivery. I would suggest to consult colleagues with expertise in this topic to expand and clarify it.

4) Response:

Thank you very much for your valuable suggestion for this section. We rewrote and changed paragraph 4. Please refer to lines:559-568, 769-793, 810-869

Author Response

Query 1: I thank the Reviewers for considering my comments. However, I think that my comments were not clear because they have not been fully responded to. New Reviewer comments are indicated by '>'. I invite the Reviewers to think a little bit more about these issues, since the present responses and changes are insufficient.

1) Response:

We thank you the Reviewer for its comments and we apologized for the insufficient answers.

Query 2: > Apologies. The point of my comment is that the Review seems to be dedicated to Tumor Microenvironment, yet this is not mentioned in the title. If you do not think its necessary, then please keep present title.

2) Response:

We thank the Reviewer for the suggestion. We propose this new title: “Non-coding-RNAs: foes or friends for targeting tumor microenvironment”.

Query 3: >The point was, the assumption is frequently made that differential expression indicates functional disease genes. Yet the goodness of this assumption should be challenged more frequently. How many changing genes are actually causally implicated in disease and are valuable therapeutic targets? Are researchers wasting time and effort blindly following this assumption.

  • Response:

We thank the Reviewer for revising our manuscript. Please refer to lines: 397-403, and 415-421, highlighted in yellow.

Query 4: > The point was, what normal cell sample should be used? If a tumor originates from a single stem cell in a complex tissue mixture comprising a heterogeneous mixture of cell types, then the bulk tumor sample (which originated from that stem cell) is likely to have very different gene expression characteristics from the surrounding tissue, independently of its tumor status. Researchers rarely consider this issue. It is somewhat unavoidable, but could perhaps be mitigated by single cell approaches.

4) Response:

We thank you the Reviewer for the suggestions. The main subject of this review is not RNA-sequencing technology in terms of its applications, and challenges for the researchers.  We strongly believe that focusing more on the issues of the single cell analysis will shift on the different subject the present review.

Query 5: > Bulk RNAseq averages across all cells, particularly overlooking rare but possibly important cell types (including originator cells of the tumour possibly).

5) Response:

We thank you the Reviewer for the comment. Please refer to lines: 403-411, highlighted in yellow.

Query 6: > Again, quite a superficial response. How do those somatic mutations affect microRNAs to promote tumor growth? What about germline variants? There is a publication from Calin's group on a germline variant (Redis et al).

6) Response:

Thank you very much for your valuable suggestion for this section. We rewrote a paragraph regarding germline and somatic mutations in miRNA genes using George Adrian Calin, Carlo Maria Croce; MicroRNA-Cancer Connection: The Beginning of a New Tale. Cancer Res 1 August 2006; 66 (15): 7390–7394. https://doi.org/10.1158/0008-5472.CAN-06-0800 and Ziebarth, J.D.; Bhattacharya, A.; Cui, Y. Integrative Analysis of Somatic Mutations Altering MicroRNA Targeting in Cancer Genomes. PLOS ONE 2012, 7, e47137, doi:10.1371/journal.pone.0047137. Please refer to lines: 100-113, highlighted in yellow.

Query 7: > Thank you for making these changes. However, in the section on Gapmers, the example provided concerns a protein coding gene (GGCT). Since there are so many studies targeting ncRNAs in cancer using Gapmers, I am puzzled why a protein coding example was chosen. Nice cases to use could be MALAT1 (Gutschner) or SAMMSON (Leucci).

7) Response:  

We thank you the Reviewer for comments and we apologized for the insufficient answer for this query. We rewrote a paragraph adding table about tremendous role of Gapmers in anticancer therapy. Please refer to lines: 871-890 and Table 3, highlighted in yellow.

We appreciate the opportunity and hope that the manuscript is now acceptable in its present form.